# Clinical code usage in UK general practice: a cohort study exploring 18 conditions over 14 years

Salwa S Zghebi ,[1,2] David Reeves ,[1,3] Christos Grigoroglou,[4] Brian McMillan ,[1,2] Darren M Ashcroft,[1,5] Rosa Parisi,[1,6] Evangelos Kontopantelis [1,6]

For numbered affiliations see end of article.

**Correspondence to**
Dr Salwa S Zghebi;
salwa.zghebi@manchester.ac.uk

## ABSTRACT

**Objective** To assess the diagnostic Read code usage for 18 conditions by examining their frequency and diversity in UK primary care between 2000 and 2013.

**Design** Population-based cohort study

**Setting** 684 UK general practices contributing data to the Clinical Practice Research Datalink (CPRD) GOLD.

**Participants** Patients with clinical codes for at least one of asthma, chronic obstructive pulmonary disease, diabetes, hypertension (HT), coronary heart disease, atrial fibrillation (AF), heart failure, stroke, hypothyroidism, chronic kidney disease, learning disability (LD), depression, dementia, epilepsy, severe mental illness (SMI), osteoarthritis, osteoporosis and cancer.

**Primary and secondary outcome measures** For the frequency ranking of clinical codes, canonical correlation analysis was applied to correlations of clinical code usage of 1, 3 and 5 years. Three measures of diversity (Shannon entropy index of diversity, richness and evenness) were used to quantify changes in incident and total clinical codes.

**Results** Overall, all examined conditions, except LD, showed positive monotonic correlation. HT, hypothyroidism, osteoarthritis and SMI codes' usage had high 5-year correlation. The codes' usage diversity remained stable overall throughout the study period. Cancer, diabetes and SMI had the highest richness (code lists need time to define) unlike AF, hypothyroidism and LD. SMI (high richness) and hypothyroidism (low richness) can last for 5 years, whereas cancer and diabetes (high richness) and LD (low richness) only last for 2 years.

**Conclusions** This is an under-reported research area and the findings suggest the codes' usage diversity for most conditions remained overall stable throughout the study period. Generated mental health code lists can last for a long time unlike cardiometabolic conditions and cancer. Adopting more consistent and less diverse coding would help improve data quality in primary care. Future research is needed following the transfer to the Systematised Nomenclature of Medicine Clinical Terms (SNOMED CT) coding.

## INTRODUCTION

The use of electronic health records (EHRs) has increased rapidly over the last three decades.[1] This has enabled researchers from

### STRENGTHS AND LIMITATIONS OF THIS STUDY

⇒ Our study presents a contemporary longitudinal analysis of clinical code usage in UK primary care, addressing an under-reported research area.

⇒ Our findings are relevant to clinical practice as we examined 18 physical and mental conditions as recorded in primary care over 14 years, using data from a large nationally representative database.

⇒ Given the design of the recorded electronic health records, we may have missed some patients with these 18 conditions (such as patients not registered with general practices), which may have affected the observed patterns of clinical code usage.

⇒ Our analysis used Clinical Practice Research Datalink GOLD data, which are obtained from clinical practices with the Vision clinical system, and Egton Medical Information Systems (EMIS) and SystmOne practices will be using somewhat different diagnostic codes.

various disciplines to examine cross-sectional and longitudinal trends of large population medical records to address many clinical research questions. EHRs are increasingly used for clinical management, clinical audits and research with real-world data, applying cross-sectional to longitudinal study designs to address descriptive epidemiology, pharmacoepidemiology, interventions evaluation and risk prediction modelling.[2 3] The available routinely collected data are far from perfect, but they provide a wealth of high-quality information on patients' clinical conditions, referrals and medication usage,[4] informing important components of clinical practice such as clinical decision-making.

Since the beginning of medical computing systems usage from the early 1970s,[5 6] the UK's primary care systems became fully computerised by 2003.[7 8] This transition was facilitated by Read codes, a comprehensive computerised semihierarchical clinical classification system designed for use in EHRs, which are

still in use in the UK.[9] These were originally developed by a clinician, Dr James Read, in the early 1980s and became the main coding system for clinical data in the UK from the mid-1990s, succeeding the Oxford Medical Information System (OXMIS) codes that were the most widely used system throughout the 1980s.[10–12] However, the Systematised Nomenclature of Medicine Clinical Terms (SNOMED CT), a systematically organised collection of medical terms, is being rolled out in general practices in a phased approach from April 2018 to replace Read codes, and it includes symptoms, diagnoses, procedures, family history, allergies and devices.[13 14] With evident increasing complexity of most healthcare disciplines,[15] such clinical terminologies make collated patient records more manageable in clinical practice settings.[16 17] To support users, national standards and guidelines are available on the use of clinical coding.[14 18] Several UK primary care electronic databases exist and are managed by different and varying computer software systems (EMIS, Vision and SystmOne), with Read codes still being the most common system through which to capture primary care clinical information. In the UK, the largest primary care databases available for research purposes include the Clinical Practice Research Datalink (CPRD), The Health Improvement Network (THIN), ResearchOne and QResearch.[28]

Despite the fact that clinical coding is a key point in the daily functionality of routine clinical practice, studies investigating their usage in real-world electronic databases are limited, although there is an observed variation in coding practice between clinicians.[19] The use of codes is a fundamental aspect of analyses of EHRs, involving a considerable amount of work, through which researchers extract a final dataset to analyse. Clinical codes are commonly used and disseminated in the form of code lists which are compiled according to the purpose, such as diagnostic codes or family history codes (online supplemental table S1). Accurate (high specificity and sensitivity) code lists are imperative in obtaining reliable data on exposures, covariates and outcomes. Previous systematic reviews have reported overall high accuracy of discharge coding (completed by clinical and/or administrative staff) in UK EHRs data that is improving over time, wherein one review accuracy was defined as the agreement between the codes allocated after independently assessing clinical notes (acting as a 'gold' standard) and those recorded on EHRs.[20 21] However, clinical practice changes over time, at varying degrees for different conditions, which is reflected in coding practice with new codes being introduced and others made redundant.

Thus, examining and quantifying the changes in clinical code usage over time is important, since alterations in usage that have not been considered, can have important implications for the analysis of EHRs and resource allocation, and may inform public health policy. An example for EHR analysis implication, the use of a 2-year-old code list for a given medical condition, may or may not be a problem, depending on how much clinical practice has changed over time for that condition. This change in

clinical practice may be driven by policy changes, such as better reimbursement for keeping a register of certain conditions. A study examining the variation in clinical code use in UK primary care using six clinical terms found that searches for the same clinical term across four different computer systems resulted in different results, for example, the mean number of codes per list ranging between 12.7 and 35.2 codes.[22] This highlighted the need for a more consistent system of code usage, with a recommendation to replace primary care code lists with shorter lists and fewer number of coding choices.[22] Importantly, the UK National Health Service (NHS) introduced the quality and outcomes framework (QOF) in April 2004, a voluntary reward and incentive programme to reward UK general practices providing high-quality care based on a range of evidence-based clinical indicators, for example, management of common chronic conditions such as diabetes and asthma.[23–25] Furthermore, important revisions were introduced to QOF in April 2006 (covering up to March 2007)[26 27] including adding new indicators for diabetes, amending diabetes clinical indicator sets and redefining the diabetes register so general practitioners (GPs) required to identify patients with diabetes as either having type 1 or type 2 diabetes, which have potentially increased the capture of diabetes cases on that period. In this study, we used data from the UK CPRD GOLD database to examine the (1) frequency ranking of diagnostic clinical codes for 18 physical and mental health conditions and (2) changes in the usage of individual clinical codes (incident vs total codes) for these conditions between 2000 and 2013 covering the period before and after the launch of the QOF.

## METHODS
### Data source and study design
We used data from the GOLD database of the UK CPRD, which comprises data from contributing anonymised general practices using the Vision clinical computer system.[28] The CPRD is one of the world's largest longitudinal electronic medical databases providing anonymised data from primary care, and is broadly representative of the UK population.[8 29] The CPRD is structured to provide data on clinical information, referrals, consultations, immunisation, tests and prescribed therapies. Up to July 2013, the CPRD held data for 11.3 million patients registered in 674 general practices. Of these, 4.4 million were active patients (representing 6.9% of the total UK population), and 6.9 million records represent inactive patients (people who have died or are no longer registered with a participating general practice).[29]

Using financial year intervals between 1 April 2000 and 31 March 2013, we examined the changes in the use of diagnostic clinical codes for 18 exemplar medical conditions in UK practices: asthma, chronic obstructive pulmonary disease (COPD); diabetes mellitus (DM), both types; hypertension (HT); coronary heart disease (CHD); atrial fibrillation (AF); heart failure (HF); stroke,

hypothyroidism, chronic kidney disease (CKD), learning disability (LD), depression, dementia, epilepsy, severe mental illness (SMI), osteoarthritis, osteoporosis and cancer. The diabetes codes included those with complications if clearly linked to diabetes, such as 'type 2 diabetes mellitus with nephropathy' (online supplemental table S1). The selected conditions, apart from osteoarthritis, were included in the QOF scheme from 2004, whereas AF, CKD, dementia, depression and LD were incentivised from 2006, and osteoporosis was incentivised from 2012. This allowed us to examine and compare QOF conditions (incentivised at different stages) plus a condition not part of the QOF (osteoarthritis).

The clinical codes used to define the examined conditions are listed in the ClinicalCodes online repository.[30] Each condition was examined as an incident code (using codes to identify new cases) and total codes (incident and prevalent cases) for each year during the study period.

### Data analysis

To examine the consistency of clinical code use across time, we applied canonical correlation analysis (CCA)[31 32] to estimate 1-year (eg, 2006–2007), 3-year (eg, 2006–2009), and 5-year canonical correlations (CCs) (eg, 2006–2011) for code usage for each of the 18 conditions based on ranking the percentage frequency use of codes. CCA is a descriptive multivariable method that provides a measure of the CC between two groups of variables or two data matrices that should be numerically complete and non-missing. CCA finds the best linear combinations maximising the correlation ($\gamma_1$) between $p$ variables in group one and $q$ variables in group 2, where the variables are measured across a common set of units (eg, general practices)[33]:

$$Y^1 = \left( Y_1^1, \cdots, Y_p^1 \right) \qquad (1)$$

$$Y^2 = \left( Y_1^2, \cdots, Y_q^2 \right), \qquad (2)$$

where $Y^1$ represents the set of $p$ outcomes in group 1, and $Y^2$ represents the set of $q$ outcomes in group 2. Consider the two linear combinations $\alpha'Y^1$ and $b'Y^2$, where $\alpha'$ is a $p \times 1$ vector of weighting coefficients and $b'$ is likewise a $q \times 1$ vector; the CC ($\gamma_1$) is given by the choice of $\alpha'$ and $b'$ that maximises the correlation between $\alpha'Y^1$ and $b'Y^2$[33]:

$$\gamma_1 = \max_{a,b} Corr\left( a'Y^1, \; b'Y^2 \right) = \max_{a,b} \frac{a'\Sigma_{12}b}{\sqrt{a'\Sigma_{11}a \, b'\Sigma_{22}b}}, \qquad (3)$$

where

$$\sum_{11} = Cov\left( Y^{(1)}, Y^{(1)} \right); \; \sum_{12} = Cov\left( Y^{(1)}, Y^{(2)} \right); \text{and}$$
$$\sum_{22} = Cov\left( Y^{(2)}, Y^{(2)} \right) \qquad (4)$$

In the present study, for a given practice, the $Y$s represent the relative use of each clinical code for a particular condition, expressed as a percentage of the total use across all codes for that condition. For example, for the 2006–2007 year-on-year diabetes correlation, the $Y$s represent the relative use of each diabetes code expressed as a percentage of the total use across all diabetes codes, where group 1 (represented by $Y^1$) would be the percentage frequency use of each clinical code for diabetes recorded in year 2006, whereas group 2 (represented by $Y^2$) would be the corresponding percentage frequency use for each corresponding diabetes code recorded in year 2007, at the general practice level. The same applies for the correlations of 3 and 5 years.

We analysed percentage frequencies rather than frequency counts so as to remove any effects of variations in practice size or disease prevalence from the estimated CCs. CCs were calculated using the R statistical software ccaPP package[34] with the 'Spearman' method, by which the weighted linear combinations $\alpha'Y^1$ and $b'Y^2$ for each year are ranked across practices prior to computation of the correlation. This method produces estimates that are more robust against model mis-specification.[35]

Numbers of incident clinical codes could be small for some conditions and practices, which can lead to biased estimates of the CCs. To adjust for this, we applied the Jackknife bias correction to the estimation of CCs for the incidence of clinical codes.[33]

For each of the 18 conditions, we also quantified changes in incident and total clinical code usage applying three measures of diversity. First, the Shannon entropy ($H$), an equitability and popular index of diversity. The index is interchangeably referred to as Shannon entropy or Shannon index where the term 'entropy' indicates the uncertainty or variability of information in a variable whose diversity is assessed by the Shannon index. The Shannon entropy index ($H$) was calculated as

$$H = -\sum_i \left( p_i \, ln \, p_i \right)$$

where $p_i$ is the proportion of a clinical code $i$ usage in a given year.

Second, we examined the richness ($S$) of clinical code usage by calculating the annual total number of incident and all codes used in a given year. Third, we estimated the evenness ($J$) of incident and total codes' usage, a measure of the relative usage of codes within a given year. In other words, evenness will be high if all codes have a similar distribution (eg, 100 diabetes records based on using four different diabetes codes, 25 times each), whereas it will be low if a few codes dominate the code usage (eg, 100 diabetes records based on using one code 70 times and another code 30 times). $J$ ranges between 0 and 1, with $J = 0$ indicating no evenness and $J = 1$ indicating complete evenness. Evenness was calculated annually by dividing Shannon index ($H$) over the natural logarithm of richness ($S$):

$$J = \frac{H}{ln(S)}$$

To simplify what these diversity measures imply, we describe a hypothetical example: if diabetes was represented using three diagnostic codes: code A (used 100

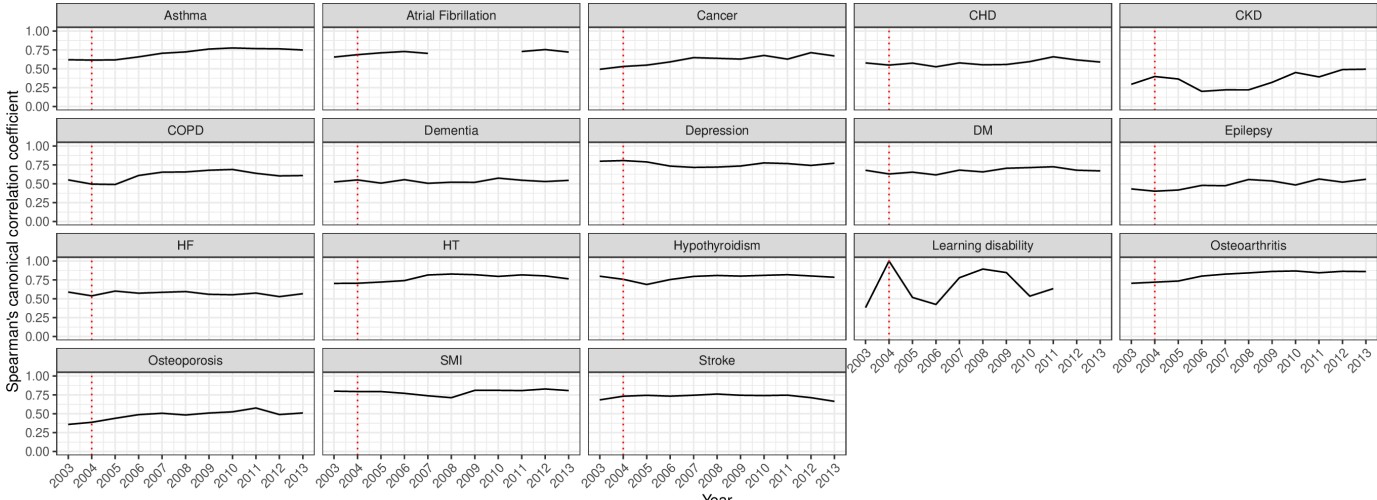

**Figure 1** Canonical correlations using 3-year window of clinical code usage for 18 mental and physical conditions. The red line represents the launching year of the QOF in 2004. CHD, coronary heart disease; CKD, chronic kidney disease; COPD, chronic obstructive pulmonary disease; DM, diabetes mellitus; HF, heart failure; HT, hypertension; QOF, quality and outcomes framework; SMI, severe mental illness.

times), code B (used 175 times) and code C (used 350 times), then the proportions of codes would be 0.16, 0.28 and 0.56, respectively. Shannon's entropy index ($H$) will be $=-1\times((0.16\times\ln 0.16)+(0.28\times\ln 0.28)+(0.56\times\ln 0.56))=0.97$; richness ($S$)=3; and evenness ($J$)=0.97/ln (3)=0.88. All analyses were conducted using R software[36] and were visualised using the ggplots2 package. A copy of the R code is presented in online supplemental table S2.

### Patient and public involvement
No patients or members of the public were involved in this study.

### RESULTS
#### Clinical code frequency ranking
Correlation of code usage over a 3-year period showed a positive association for most conditions (figure 1). Strong, overall positive and monotonic correlation (CC >0.7) was observed for depression, HT, hypothyroidism, osteoarthritis, SMI and stroke. Positive, monotonic but weaker associations were observed for CKD, epilepsy and osteoporosis. LD showed a non-monotonic function with fluctuations ranging between 0.4 and 1.0 and a notable decline after 2004 before increasing again from 2007. The window correlations of 1 and 5 years showed similar overall trends, but the association was slightly decreasing as the window increased. Clinical conditions with the highest correlation levels were asthma, AF, cancer, CHD, depression, diabetes, HT, hypothyroidism, osteoarthritis, SMI and stroke for the 1-year window (online supplemental figure S1). For the 5-year window, HT, hypothyroidism, osteoarthritis and SMI codes' usage was overall highly correlated mainly in recent years (online supplemental figure S2). On the other hand, conditions with the lowest correlations (CC ≤0.6) were CKD and LD (for

most years) for the 1-year window, and cancer, CHD, CKD, COPD, dementia, diabetes, epilepsy, HF, LD and osteoporosis for the 5-year window.

Over a 3-year window, strong correlations for incident code usage (Jackknife bias corrected CC ≥0.6) were observed for all examined conditions except CKD, epilepsy and osteoporosis (figure 2). Similarly, the window correlations of 1 and 5 years showed similar trends but lower coefficients with longer windows (online supplemental figures S3 and S4, respectively).

#### Clinical code usage diversity
Data from 684 UK general practices contributing to the CPRD GOLD were used. Overall, the diversity indices of code usage were stable over the study period for most conditions but with wide CIs. Higher entropy ($H$) indices were observed with cancer, diabetes and SMI ($H$ between 2 and 4), while the lowest levels were observed with LD and osteoporosis ($H$ between 0 and 2) (figure 3). Over time, the entropy index of code usage remained stable for most conditions but increased gradually for asthma, COPD, diabetes, HF and osteoporosis (primarily incident codes). Fluctuations and/or a separation between the incident and total codes trends were observed around 2006, mainly for AF, dementia, depression, CKD and LD. The Shannon index ($H$) for incident codes had a similar trend to that for total codes for most conditions over time, except for cardiovascular disease (CVD) and diabetes, where it exceeded total codes.

Across the examined conditions, the richness ($S$) of incident and total code usage (number of codes used) was highest for cancer (>500 codes), diabetes and SMI (≥250 codes each) and lowest for AF, hypothyroidism and LD ($S$<100) (figure 4). The trends, however, remained stable throughout the study period, except a small decrease for SMI codes and a decrease in cancer after a

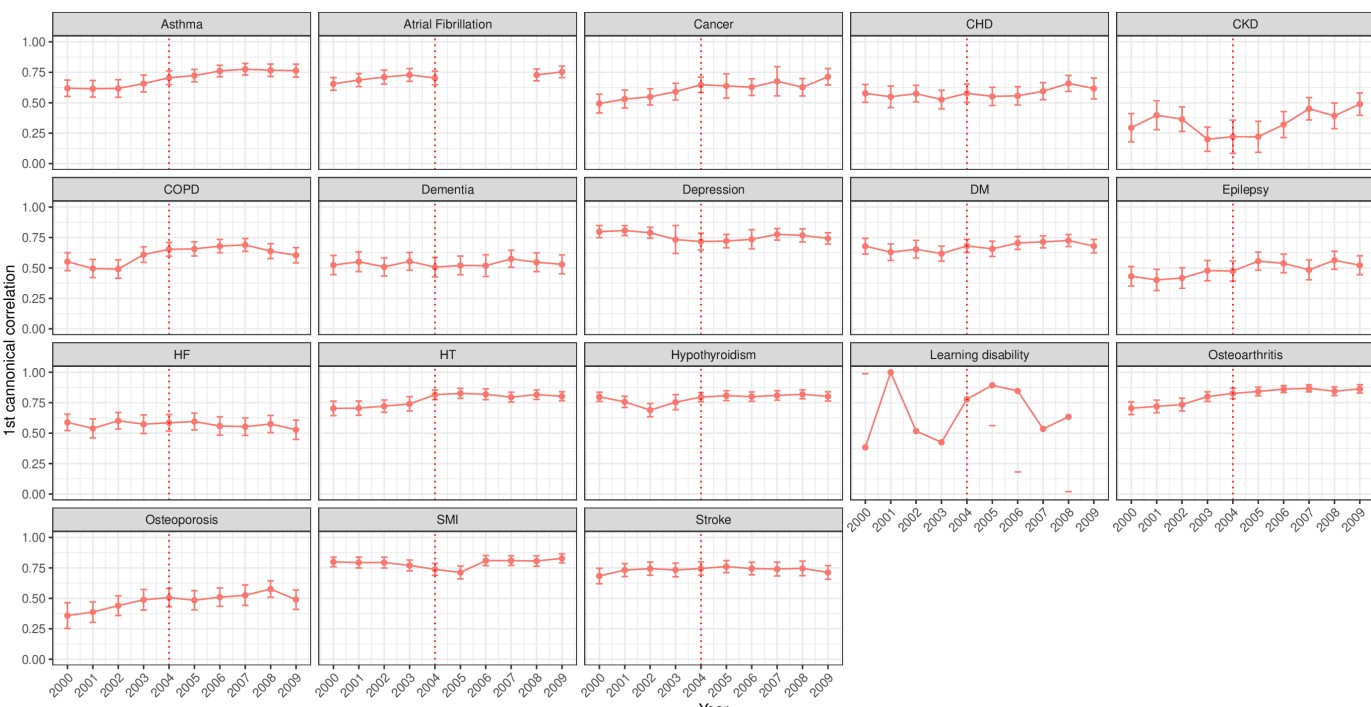

**Figure 2** Bias-corrected canonical correlations (95% CI) using 3-year window for incident clinical code usage for 18 mental and physical conditions. The incident code is a clinical code indicating new (incident) cases. The red line represents the launching year of the QOF in 2004. CHD, coronary heart disease; CKD, chronic kidney disease; COPD, chronic obstructive pulmonary disease; DM, diabetes mellitus; HF, heart failure; HT, hypertension; QOF, quality and outcomes framework; SMI, severe mental illness.

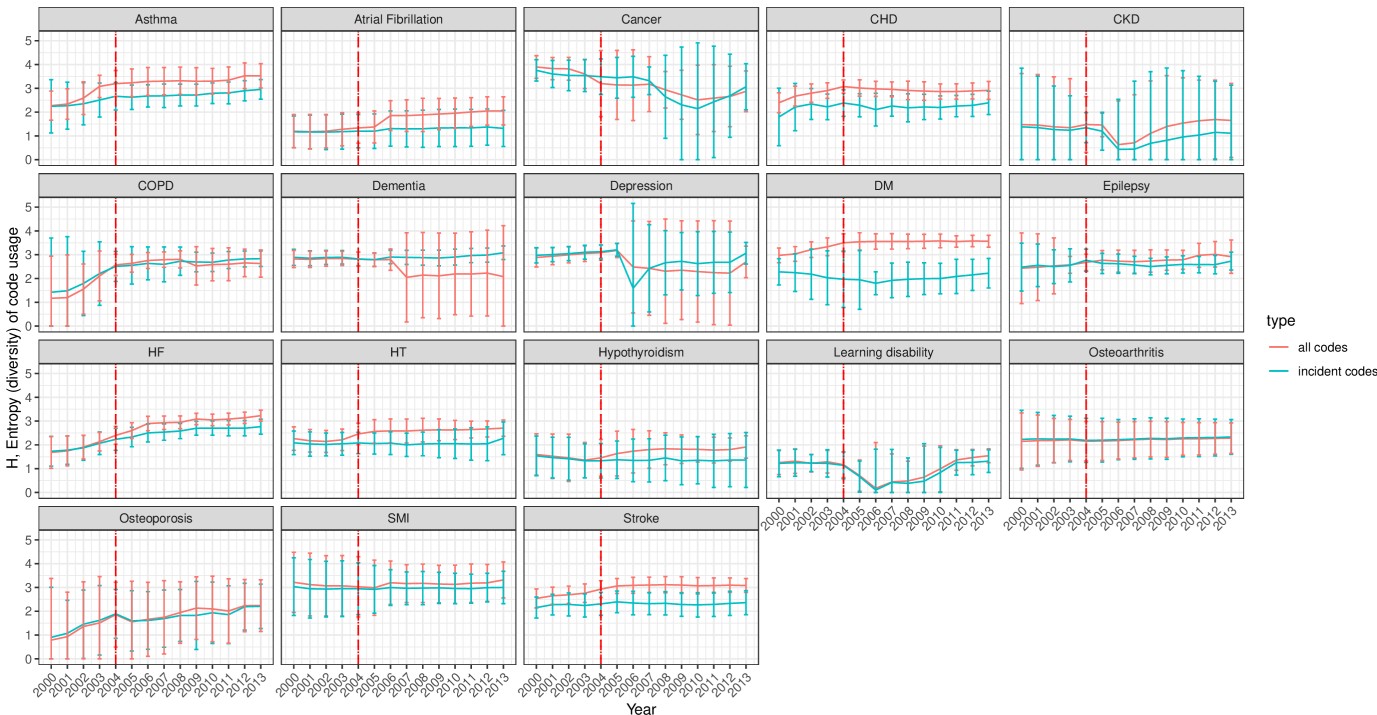

**Figure 3** Entropy (95% CI) of incident and all clinical code usage for 18 mental and physical conditions. The incident code is a clinical code indicating new (incident) cases. All codes indicate any diagnostic clinical code for the condition incident and prevalent cases. The red line represents the launching year of the QOF in 2004. The 95% CIs were calculated as the mean±1.96×SE (SE has been estimated using jackknife approach). CHD, coronary heart disease; CKD, chronic kidney access; COPD, chronic obstructive pulmonary disease; DM, diabetes mellitus; HF, heart failure; HT, hypertension; QOF, quality and outcomes framework; SMI, severe mental illness.

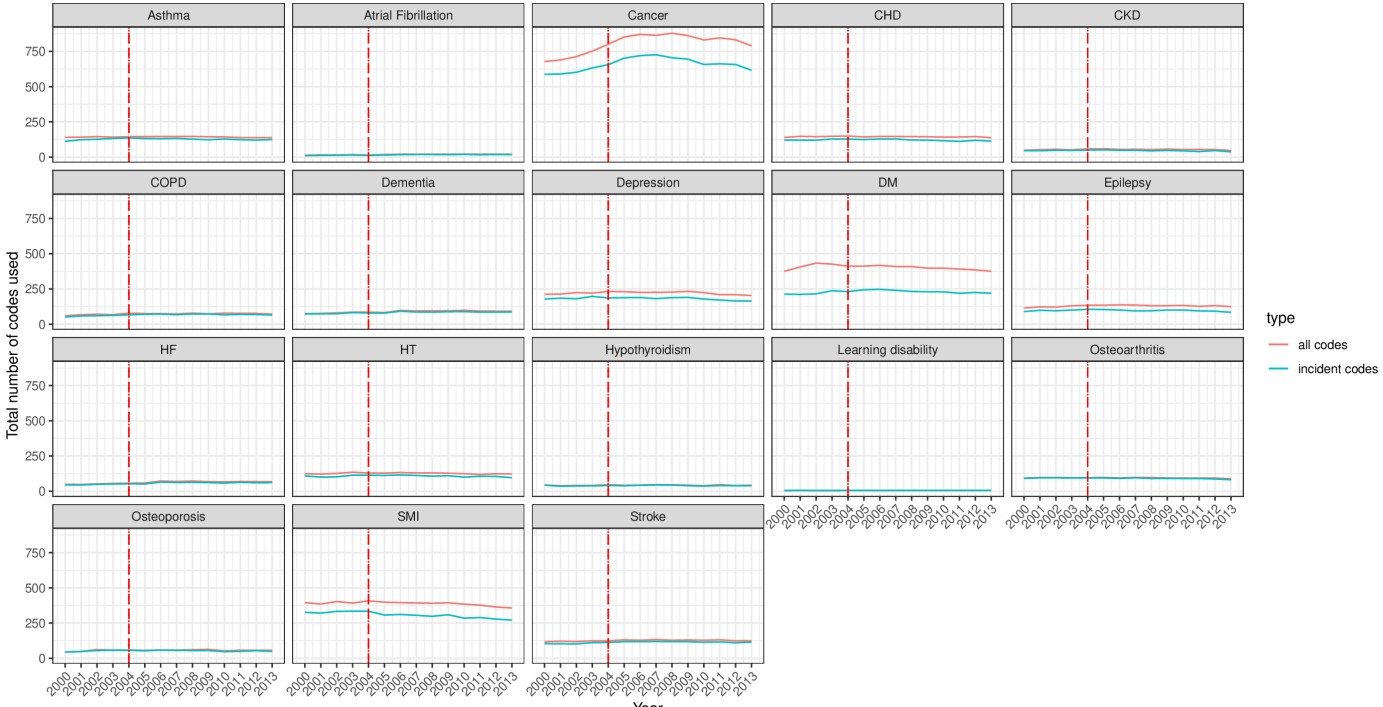

**Figure 4** Richness of incident and all clinical code usage for 18 mental and physical conditions. The incident code is a clinical code indicating new (incident) cases. All codes indicate: any diagnostic clinical code for the condition incident and prevalent cases. The red line represents the launching year of the QOF in 2004. CHD, coronary heart disease; CKD, chronic kidney disease; COPD, chronic obstructive pulmonary disease; DM, diabetes mellitus; HF, heart failure; HT, hypertension; QOF, quality and outcomes framework; SMI, severe mental illness.

brief rise between 2000 and 2005. The difference between the number of incident and total codes for SMI, diabetes and cancer was evident (total codes more than incident codes), unlike in the other conditions where the $S$ index was similar for both code categories.

The evenness ($J$) of both incident and total codes was overall stable and almost identical at least up to 2006, before total codes surpassed incident codes for most conditions except for depression and dementia, where the $J$ index for incident codes exceeded that of total codes (figure 5). The two exceptions to this observation were LD and CKD. For LD, evenness was stable at ~0.75 between 2000 and 2003, declined in 2004 before reincreasing from 2007 and returning to pre-2004 levels from 2011 onwards. For CKD, evenness dipped briefly around 2006–2007 and started to increase again from 2008 until the end of the study period (2013). Given the calculation formula, it is worth noting that the trends of entropy were similar to that of evenness for conditions with low richness, namely, for AF, dementia, HF, HT, hypothyroidism, LD, osteoarthritis and osteoporosis.

## DISCUSSION
### Main findings
We assessed the clinical code usage for 18 conditions recorded in a large nationally representative UK EHR between 2000 and 2013. The results show overall strong positive monotonic correlation for all examined conditions except LD, which showed a fluctuating pattern during the study period. The CCs diminished over longer windows (5-year vs 1-year window). HT, hypothyroidism, osteoarthritis and SMI had the highest 5-year correlation, mainly in later years of the study period.

The codes' usage entropy and evenness diversity measures remained overall stable throughout the study period for most conditions, except gradual increases over time for respiratory conditions, diabetes, HF and osteoporosis. This increase in diversity may be partially due to the regular addition of new diagnostic codes and domains over time. For example, major revisions were introduced to the QOF in April 2006, resulting in the addition of new clinical areas and indicators.[27] As a consequence, CKD is among the conditions that it has been acknowledged to have benefited from these revisions as the CKD domain was added in 2006, reflected in improved recording in primary care from that year onwards.[37] For most conditions (except LD), evenness (indicating the abundance of codes in a sample) was overall ≥0.5, suggesting a uniform distribution of the codes. Cancer, diabetes and SMI had the highest richness indices among all examined conditions.

### Comparison with previous studies
Observational studies examining variations in clinical code usage are limited. A recent study examined the code usage of CVD between 2001 and 2015 in primary and secondary care records in England.[38] The study aimed

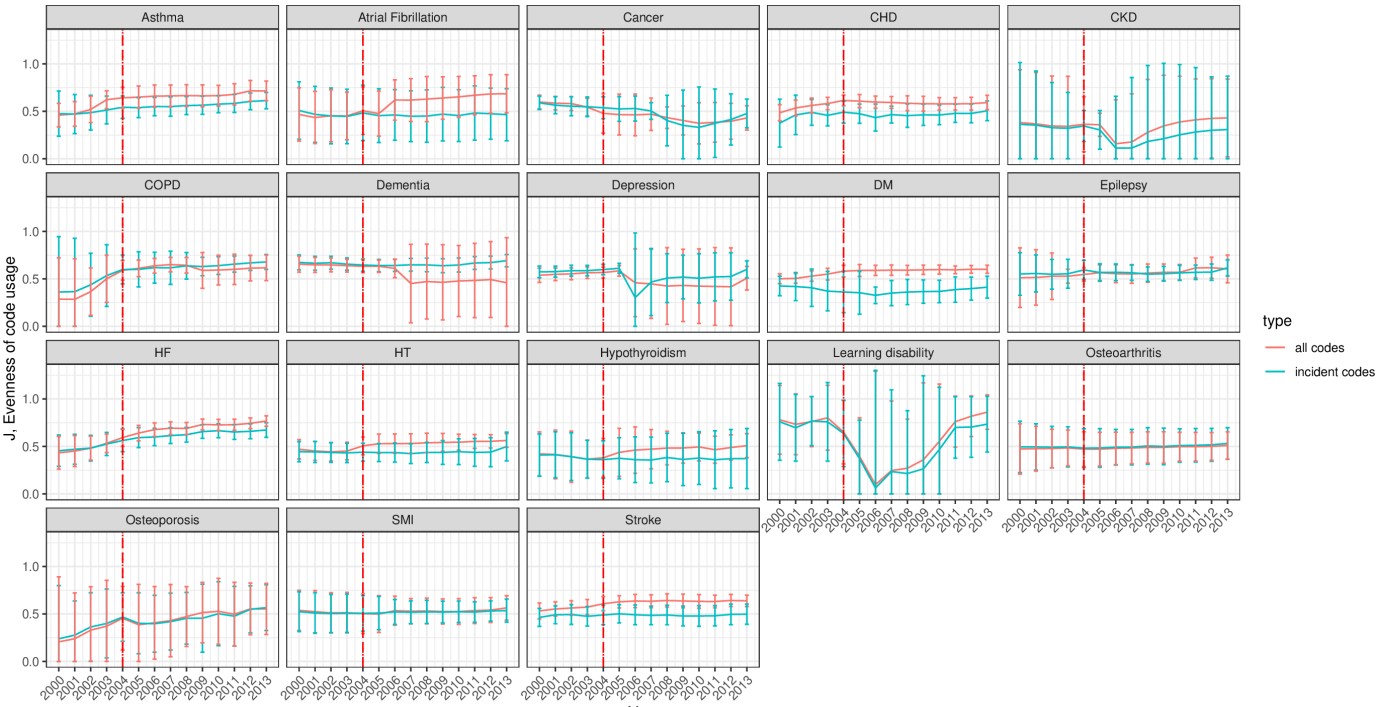

**Figure 5** Evenness (95% CI) of incident and all clinical code usage for 18 mental and physical conditions. The incident code is a clinical code indicating new (incident) cases. All codes indicate any diagnostic clinical code for the condition incident and prevalent cases. The red line represents the launching year of the QOF in 2004. The 95% CIs were calculated as the mean±1.96×SE (SE has been estimated using jackknife approach). CHD, coronary heart disease; CKD, chronic kidney disease; COPD, chronic obstructive pulmonary disease; DM, diabetes mellitus; HF, heart failure; HT, hypertension; QOF, quality and outcomes framework; SMI, severe mental illness.

to examine if temporal variability methods can identify changes in CVD recording by quantifying the differences of monthly distributions of the variables of interest: CVD status and sociodemographic variables. The study found variability in the frequency of CVD codes across time, potentially due to non-medical causes such as changes in coding used and coding guidelines, for example, changes in InternationalClassification of Diseases (ICD) coding in hospital records. Despite relevance, their approach (examining the prevalence of CVD stratified by patient demographic variables) differs from our methods and hence the results are not directly comparable. In addition, we examined code usage for 18 conditions, including CVD, from UK general practices.

A study by Tai *et al* examined the diversity of data entry screens in four clinical computer systems available in UK general practices and assessed its impact on the variation and quality of recorded clinical data for six exemplar conditions (sore throat, tired all the time, depression, cystitis, type 2 diabetes and myocardial infarction).[22] In agreement with what we reported on the large number of available codes for some conditions (high richness), Tai and colleagues found that searches for the same clinical term across the systems resulted in different results and found long code lists where the mean number of codes ranged between 12.7 and 35.2 codes per list.[22] Their study concluded that the systems may contribute towards a diverse coding in primary care, suggesting the need to

standardise clinical coding across systems and to adopt shorter and more restricted code lists to help improve data quality. This is an important issue for UK primary care, since the semistructured and dynamic nature of Read codes often results in diverse and long clinical code lists. Their findings highlight the need for analyses as ours, which are lacking in current literature, investigating the real-world abundance and trends of code usage over time derived from routine clinical data. Additionally, some general practice systems use CTV3 clinical codes and not Read codes resulting in the availability of two versions of clinical codes. SNOMED CT system, which is gradually being implemented across UK primary care from 2018, aimed to provide a single clinical terminology for effective and consistent exchange of clinical data across all NHS settings to help improve patient care and data analysis.[39] Being an international clinical terminology, SNOMED CT will allow the UK to participate in global healthcare research.

Our results showed that diabetes codes usage (types 1 and 2) had one of the highest richness index levels (number of codes used), while the diversity entropy index was steadily increasing over the study period, highlighting the increasing variety of diabetes codes used in primary care over time. This observation agrees with a previous study that examined the Read codes used to identify diabetes management in people with diabetes registered with 17 general practices in one locality in London.[11]

That study concluded that a wide range of diabetes codes were used and that the number of people assigned each code differed across practices. This again indicates that an approach is required to standardise clinical code lists and thereby coding usage as much as possible, minimise clinical recording errors and improve research robustness.

## Implication of findings

Our findings shed additional light on the use of clinical codes in research. We found that HT, hypothyroidism, osteoarthritis and SMI codes' usage are highly correlated over the 5-year window (ie, the codes' usage was similar across years), whereas cancer, CHD, CKD, COPD, dementia, diabetes, epilepsy, HF, LD and osteoporosis had the lowest correlation over the same window. In terms of clinical code lists' size required to define a condition (richness), we found that conditions with the highest richness across the study period were cancer, diabetes and SMI (between 250 and 875 codes), whereas AF, hypothyroidism and LD had the lowest richness (<100 codes). Collectively, these findings indicate that diabetes, cancer and SMI codes have high richness and need to be defined carefully and then they can either last for 5 years (SMI) or only 2 years (diabetes and cancer), whereas hypothyroidism has low code usage richness and can last for 5 years. This might be due to that diabetes is often a target of government initiatives, unlike hypothyroidism, which is rarely a focus of such interventions.

The results suggest that defining cohorts of people with mental health conditions (SMI and depression) over time was less sensitive to the changes of code usage (up to 5 years old) compared with most cardiometabolic conditions and cancer.

The observed findings also suggest the need to adopt a more consistent and less diverse coding in primary care, as this will help improve data quality. Inconsistent use of clinical coding may result in people with the same condition not being flagged as having the condition,[19] which may have implications on searching and identifying these people for clinical and research purposes, or to identify people for shielding measures or those who are a priority for a vaccination as in the current COVID-19 pandemic. While acknowledging that SNOMED CT is gradually replacing Read codes in general practice care since April 2018, our findings are still relevant in documenting the clinical code usage over a long period where Read codes were the main UK coding system. There is potential for SNOMED CT terminology to improve coding consistency, mainly through the plan to implement it in both primary care and secondary care systems.[40] However, a possible issue with SNOMED terminology is the need for specialist browser and reference sets, such as the general practice reference set to handle the long hierarchies of SNOMED system.[40] A reference set is a mechanism that can be employed to represent value sets of SNOMED CT components.[41]

Also, the rapidly increasing complexity of healthcare systems[15] might play a role on the observed trends in code usage over time. In other words, code usage practices (eg, the tendency of data enterer to use easily accessed and well-known codes) may be partially driven by personal and work factors in the complex healthcare systems, such as limited time and organisational factors. A possible relationship between clinical code selection and epidemiology of chronic conditions has been reported previously, for example, for diabetes.[19 42] From GPs' stance, one thing that may have changed in recent years is the coding of people being at 'high risk of DM', and it is something that GPs are increasingly aware of (ie, people with glycated haemoglobin $A_{1c}$ ($HbA_{1c}$ 42–47 mmol/mol).

## Strengths and limitations of the study

Our study has several strengths. Using a range of frequency and diversity measures, we present a contemporary longitudinal analysis of clinical code usage in UK primary care, while only a few existing studies have addressed this research area. We used data from a large nationally representative database, where the validity of recorded diagnostic coding has been acknowledged previously.[43] Additionally, the data quality is assumed to be high, as it is based on QOF clinical code lists (except osteoarthritis). Our findings are relevant to clinical practice as we examined a broad range of prevalent physical and mental illnesses as recorded in primary care and considered the clinical implications of variations in clinical coding over 14 years.

Our study has also several limitations. Given the design of the recorded EHRs, we may have missed some patients with the examined conditions due to some unusual circumstances or settings, such as patients not registered with general practices (eg, homeless people), which may have affected the observed patterns of clinical code usage. Also, analyses were not extended to examine ICD, 10th Revision (ICD-10), clinical codes in secondary care setting (only available in England), as our aim was to focus on the usage of Read codes recorded in UK primary care visits as the main point of clinical care. CCA provides a single multivariate measure of correlation, thus simplifying interpretation compared with analysing each clinical code separately. However, the measure represents the maximum possible correlation between frequencies of code use at two different time points and does not account for the code set being the same at both times and hence may over-represent actual agreement to a degree. This is an intrinsic limitation of CCA is that it does not consider the 'code' per se but its frequency, so it would return high correlation for possible scenarios such as if a code merges at a time point with another code into a single code or if, hypothetically, all practices transition from one code to another at the same time. As CCA is based on the correlation of two positive-definite matrices of data that should be numerically complete, problematic quality and levels of recording as observed with AF and LD recording on some time points results in missing values as observed in, for example, figures 1 and 2 and online supplemental figure S4. In addition, although the clinical relevance of the examined conditions which are also prevalent outside UK, such as diabetes, CVD and cancer, and that our findings highlight

the need of consistent coding lists applies for non-UK national health system with established or aiming to develop electronic clinical coding, our study may have limited generalisability to non-UK systems. Finally, we used CPRD GOLD, which collects data from general practices using the Vision clinical system, and code usage will vary to some extent in general practices using EMIS or SystmOne. However, we would expect such variation to be low in chronic conditions incentivised through the QOF, with specific common code lists used by practices to ensure remuneration eligibility.

## CONCLUSIONS

The code usage in UK primary care was overall stable for most of the examined chronic conditions managed in general practice between 2000 and 2013, but, as would be expected, the changes were higher over longer time windows. Diabetes, cancer and SMI code lists have high richness and need to be defined carefully by researchers and/or clinicians, which might be considerably time-consuming, but once defined, SMI codes can last up to 5 years, while diabetes and cancer codes can last for only 2 years. On the other hand, hypothyroidism has low richness but also can last up for 5 years. Our study addresses an under-reported research area, and the findings suggest the need to adopt a more consistent and less diverse coding in primary care to help improve data quality and the use of recent codes for cardiometabolic conditions and cancer. More research is needed in this area following the full transfer to the SNOMED CT coding and to examine the code usage in secondary care settings.

**Author affiliations**

[1]NIHR School for Primary Care Research, Centre for Primary Care and Health Services Research, Manchester Academic Health Science Centre (MAHSC), The University of Manchester, Manchester, UK
[2]Division of Population Health, Health Services Research and Primary Care, School of Health Sciences, Faculty of Biology, Medicine and Health, Manchester Academic Health Science Centre (MAHSC), The University of Manchester, Manchester, UK
[3]Centre for Biostatistics, School of Health Sciences, Faculty of Biology, Medicine and Health, Manchester Academic Health Science Centre (MAHSC), The University of Manchester, Manchester, UK
[4]Manchester Centre for Health Economics, Division of Population Health, Health Services Research and Primary Care, Manchester Academic Health Science Centre (MAHSC), The University of Manchester, Manchester, UK
[5]Centre for Pharmacoepidemiology and Drug Safety, School of Health Sciences, Faculty of Biology, Medicine and Health, Manchester Academic Health Science Centre (MAHSC), The University of Manchester, Manchester, UK
[6]Division of Informatics, Imaging, and Data Sciences, School of Health Sciences, Faculty of Biology, Medicine and Health, Manchester Academic Health Science Centre (MAHSC), The University of Manchester, Manchester, UK

**Acknowledgements** The authors thank Dr David A Springate (DAS) for extracting and analysing the data.

**Contributors** EK and DAS designed the study. DAS extracted the data from all sources and performed the initial analyses. RP, DR, and SSZ validated the analyses; RP developed the final figures. SSZ wrote the manuscript and EK, RP, DR and CG critically edited the initial drafts; DMA and BM contributed to interpretation of data and revised the paper for important intellectual content. All authors agreed on the final version of the paper before submission. SSZ is the guarantor of this work and, as such, had full access to all the data in the study and takes responsibility for the integrity of the data and the accuracy of the data analysis.

**Funding** This study is funded by the National Institute for Health and Care Research (NIHR) School for Primary Care Research (grant number 211). The views expressed are those of the authors and not necessarily those of the NIHR or the Department of Health and Social Care.

**Competing interests** DMA reports research grants from Abbvie, Almirall, Celgene, Eli Lilly, Novartis, UCB and the Leo Foundation.

**Patient and public involvement** Patients and/or the public were not involved in the design, conduct, reporting or dissemination plans of this research.

**Patient consent for publication** Not applicable.

**Ethics approval** This study is based on data from Clinical Practice Research Datalink (CPRD) obtained under licence from the UK Medicines and Healthcare products Regulatory Agency. The study was approved by the Independent Scientific Advisory Committee (ISAC) for MHRA Database Research (protocol number: 16_115). The data are provided by patients and collected by the NHS as part of their care and support. Generic ethical approval for observational research using CPRD with approval from ISAC has been granted by a Health Research Authority Research Ethics Committee (East Midlands—Derby, REC reference number 05/MRE04/87).

**Provenance and peer review** Not commissioned; externally peer reviewed.

**Data availability statement** Data may be obtained from a third party and are not publicly available. All data relevant to the study are included in the article or uploaded as supplementary information. Clinical code lists are available from clinicalcodes.org. Electronic health records are, by definition, considered sensitive data in the UK by the Data Protection Act and cannot be shared via public deposition because of information governance restriction in place to protect patient confidentiality. Access to data is available only once approval has been obtained through the individual constituent entities controlling access to the data. The data can be requested via application to the Clinical Practice Research Datalink.

**ORCID iDs**
Salwa S Zghebi http://orcid.org/0000-0002-7978-1094
David Reeves http://orcid.org/0000-0001-6377-6859
Brian McMillan http://orcid.org/0000-0002-0683-3877
Evangelos Kontopantelis http://orcid.org/0000-0001-6450-5815

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
