## [Reviewer comments · BMJ Open]

ARTICLE DETAILS

TITLE (PROVISIONAL)	Clinical code usage in UK general practice: a cohort study exploring 18 conditions over 14 years
AUTHORS	Zghebi, Salwa; Reeves, David; Grigoroglou, Christos; McMillan, Brian; Ashcroft, Darren; Parisi, Rosa; Kontopantelis, Evangelos

VERSION 1 – REVIEW

REVIEWER	Filmann, Natalie Institute of Biostatistics and Mathematical Modeling, Goethe-University, Frankfurt/Main
REVIEW RETURNED	31-May-2021

GENERAL COMMENTS	The authors present a well-conducted study analysing diagnostic Read code usage for 18 conditions by examining their frequency and diversity in UK primary care between 2000 and 2013. As I am a biostatistician I mainly reviewed the statistical analysis of this work. The methods used adequately are well-described, which is especially important here, because they are non-standard methods used in clinical papers. My only concerns relate to the relevance of the topic, as it seems to be mainly uk-specific.
--

REVIEWER	Rockenschaub, Patrick University College London, Institute of Health Informatics
REVIEW RETURNED	The authors cite a paper authored by me. 09-Jul-2021

GENERAL COMMENTS	1. The authors conclude that “code usage in UK primary care was overall stable for most examined codes”. I understand that the authors come to this conclusion by independently calculating the canonical correlation for a pair of 1/3/5 year lagged set of relative codes at each practice. This quantifies the maximum correlation that can be achieved through a linear combination of the code proportions in each set. However, I would challenge that this necessarily measures coding stability. Instead, one may argue that it measures whether coding changes dissimilarly across practices. To give an example of change that would not be picked up, if two codes C1 and C2 are consolidated into a single code C1* and C2 ceases to be used, this would constitute a major shift in coding behaviour but would still result in a strong correlation measure if $p(C1^*) = p(C1) + p(C2)$ at each practice, i.e., in sum the new code is used as often at each practice as the two individual previous codes were. Could the authors please comment (ideally within the manuscript) on which changes their method is likely to pick up, and whether there are other examples like the above that the method is unlikely to identify?
--

	2. The upper boundary for the Shannon index H is proportional to the number of probabilities. While the maximum possible value for two codes with relative frequency p_1 and p_2 is 0.69 (when each = 0.5), the maximum possible value for 100 codes with p_1, \dots, p_{100} is 4.60 (when each = 0.01). This agrees with the reported findings, where the conditions with the largest number of codes (cancer, diabetes, SMI) also have the highest entropy. Evenness seems to account for this issue, showing similar trends but at a scale that is independent of the number of codes. Could the authors please comment on the added value of reporting and plotting H, or whether it would be sufficient (and simpler) to only report richness and evenness? 3. Figures 3 and 5: The lower boundary for the Shannon index H and evenness is 0. Could the authors please report how they calculated the 95% confidence intervals (which extend to <0 e.g. for COPD or learning disability) and could they adjust them to remain within the possible range (i.e., ≥ 0). 4. Figure 4: all other plots show yearly estimates. Could the authors please specify on which time scale Figure 4 was measured, as it seems to be more granular than yearly measures? Is such granularity needed or would it preferable to use the same time scale as other plots in order to aid comparability? 5. The figures are currently very inconsistent. Could the authors please strive to make them as uniform as possible, especially were figures report the same results just for different subsets or parameters. For example: a. I understand that Figures 1 (all codes) and Figure 2 (incidence codes) show the same general results but for different code sets. Could the authors harmonize/standardise these plots by using the same layout, and also report bias-corrected results for Figure 1. This would simplify comparison of these plots. b. Figure 1: Could the authors please extend the y-axis to 0 as in Figure 2? c. Figure 1: The heading “year-on-year correlations” seems misleading for 3-year correlations. Could the authors please change the heading similar to that in supplementary Figure S2? d. Figure 1: panel headers still contain underscores and all lower case. I would suggest to change headings in line with Figure 2. e. Figures 2 and S2-4: unlike Figure 1 and Figure S1, there are no red lines. Could the authors please add those lines? 6. In Figure 2 there are missing values for atrial fibrillation. Could the authors please comment on why those were observed. 7. I would invite authors to publish their analysis code online in a repository like GitHub and link to it within their manuscript. This would greatly aid reproducibility of their results.
--	---

REVIEWER	Letrilliart, Laurent Département de médecine générale, Univ. Lyon, Université Claude Bernard Lyon 1, Université Saint-Étienne, Collège universitaire de médecine générale, F-69008 Lyon, F-42023 Saint-Étienne I am an executive member of the Wonca Classification Committee
REVIEW RETURNED	11-Jul-2021

GENERAL COMMENTS	General comments This manuscript reports on a study exploring Read code usage for 18 mostly chronic conditions in UK general practice between 2000 and 2013. This study is original, but the methodology is not presented clearly consistently enough across the paper to allow the reader to fully understand the meaning of the objectives and of the too numerous results. Specific comments Introduction - The paragraph presenting the QOF should be moved just before the statement of the study objectives and linked to the issue of clinical code usage. An example of how the introduction of a new indicator in the QOF may have influenced the use of clinical codes should be provided, especially regarding type 2 diabetes. Methods and results - Were the clinical codes under study used in the GPs' EHR for the assessment of any encounter or in the patient problem list or either? - According to the first objective, the (canonical) correlation is supposed to provide a frequency ranking, but how it does is unclear. Do the correlations measure the stability of the codes used for a particular condition? I understand that the study ranks the correlation levels of the conditions under study rather than their frequency. - Figure 2 and Figure 1 are redundant, and one of them seems enough. - It is unclear how measures of diversity can quantify changes in clinical codes usage, as stated in the second objective. - What is the added value of using 3 indicators of diversity? The respective meaning of entropy and of evenness are difficult to understand and should be better explained. - It is confusing to state that the entropy index takes into account richness and evenness, since at the end this is the evenness which is calculated as a combination of the measures of entropy and richness. - In the methods section, a possible richness measure for diabetes is 3, whereas the values presented for incident diabetes mellitus in Figure 4 are around 250. Why such an inconsistency? - What is the need to present the incident codes and the total (prevalent) codes in some of the figures? The comparison between them is not reported in the main findings. - The red dotted line in some of the figures should be defined and probably included in all figures. I guess it represents the launching year of the QOF. - Some legends are missing on the x axes in Figure 1. Discussion - The second paragraph of the subsection on the Comparison with previous studies (lines 9 to 33 on p12) should be related to the results of the study. In addition, the authors report the interpretation of the results of the study by Tai et al. without reporting the results of this study. - It would be useful to discuss the potential of a limited reference set of clinical codes such as the Snomed CT subset (https://dd4c.digital.nhs.uk/dd4c/publishedmetadatas/intid/971) to improve coding consistency in general practice when Snomed CT will be implemented. - What could be the influence of changes in the epidemiology of chronic conditions over the study period on the use of some codes? For example, the prevalence of type 2 diabetes has increased in the recent decades. Can this evolution have contributed to the changing way of coding this condition?
---

	Conclusion  - The authors conclude that “the changes were higher over longer time windows”. Was it not expected or even obvious? This should probably be formulated differently. - The opposition in the following sentence is not clear enough: “Diabetes, cancer, and SMI codes need to be defined carefully but SMI codes can last up to 5 years,...”.
--	--

VERSION 1 – AUTHOR RESPONSE

Reviewer 1

Dr. Natalie Filmann, Institute of Biostatistics and Mathematical Modeling, Goethe-University, Frankfurt/Main

Comments to the Author:

- 1. The authors present a well-conducted study analysing diagnostic Read code usage for 18 conditions by examining their frequency and diversity in UK primary care between 2000 and 2013.**

As I am a biostatistician I mainly reviewed the statistical analysis of this work. The methods were used adequately are well-described, which is especially important here, because they are non-standard methods used in clinical papers.

My only concerns relate to the relevance of the topic, as it seems to be mainly uk-specific.

Authors' response

We thank the reviewer for their positive feedback on our study and paper.

We acknowledge the criticism on generalisability to other countries. We argue that the relevance of this topic and our results to wider non-UK primary care systems stems from:

The signals we found indicating the need of consistent coding lists in any national health system relying on clinical coding in EHRs or for countries aiming to develop a new or upgrade existing coding systems.

The clinical relevance of the examined conditions which are also prevalent outside UK such as DM, CVD and cancer.

In response to this comment, we have added the following text to the Limitations section of the paper:

"In addition, although the clinical relevance of the examined conditions which are also prevalent outside UK, such as diabetes, CVD, and cancer, and that our findings highlight the need of consistent coding lists applies for non-UK national health system with established or aiming to develop electronic clinical coding, our study may have limited generalisability to non-UK systems."

Reviewer 2

Mr. Patrick Rockenschaub, University College London

Comments to the Author:

2. The authors conclude that "code usage in UK primary care was overall stable for most examined codes". I understand that the authors come to this conclusion by independently calculating the canonical correlation for a pair of 1/3/5 year lagged set of relative codes at each practice. This quantifies the maximum correlation that can be achieved through a linear combination of the code proportions in each set. However, I would challenge that this necessarily measures coding stability. Instead, one may argue that it measures whether coding changes dissimilarly across practices. To give an example of change that would not be picked up, if two codes C1 and C2 are consolidated into a single code C1* and C2 ceases to be used, this would constitute a major shift in coding behaviour but would still result in a strong correlation measure if $p(C1^*) = p(C1) + p(C2)$ at each practice, i.e., in sum the new code is used as often at each practice as the two individual previous codes were. Could the authors please comment (ideally within the manuscript) on which changes their method is likely to pick up, and whether there are other examples like the above that the method is unlikely to identify?

Authors' response

We thank the reviewer for raising this point. We made the general conclusion that *'code usage in UK primary care was overall stable for most of the examined chronic conditions'* to reflect the overall observed trends for the 18 conditions. In the paper, we had acknowledged a limitation that the CCA method may, to some degree, over-represent actual agreement between pairs of code sets as it does not account for the code set being the same at both times.

By running the potential example that the reviewer suggested in our data, that scenario indeed would not be picked up by the CCA as CCA does not consider the 'code' per se but its frequency change, so when a code is merged with another code into a single code at a time point, CCA still returns high correlation. This is an intrinsic limitation of applying CCA in this context which is now clearly highlighted in the limitations section of the manuscript.

Another possible example of undetected code changes is if, hypothetically, all practices transition from code A to code B at the same time, CCA will be high, again indicating the limitation of CCA method.

In response to this comment, we added new text to the limitations section that reads as:

"CCA provides a single multivariate measure of correlation, thus simplifying interpretation compared to analysing each clinical code separately. However, the measure represents the maximum possible correlation between frequencies of code use at two different time points and does not account for the code set being the same at both times, hence may over-represent actual agreement to a degree. This is an intrinsic limitation of CCA is that it does not consider the 'code' per se but its frequency, so it would return high correlation for possible scenarios such as if a code merges at a time point with another code into a single code, or if, hypothetically, all practices transition from one code to another at the same time."

3. The upper boundary for the Shannon index H is proportional to the number of probabilities. While the maximum possible value for two codes with relative frequency p_1 and p_2 is 0.69 (when each = 0.5), the maximum possible value for 100 codes with p_1, \dots, p_{100} is 4.60 (when each = 0.01). This agrees with the reported findings, where the conditions with the largest number of codes (cancer, diabetes, SMI) also have the highest entropy. Evenness seems to account for this issue, showing similar trends but at a scale that is independent of the number of codes. Could the authors please comment on the added value of reporting and plotting H , or whether it would be sufficient (and simpler) to only report richness and evenness?

Authors' response

We believe that reporting Shannon index H is informative as it is generally acknowledged to serve as a simple 'summary' measure which encompasses two dimensions of code usage diversity: richness and evenness. From that point of view, we believe that what the reviewer highlights is not undesirable. We appreciate no measure is perfect, but we felt all three could shed some light into what we set out to investigate.

4. Figures 3 and 5: The lower boundary for the Shannon index H and evenness is 0. Could the authors please report how they calculated the 95% confidence intervals (which extend to <0 e.g. for COPD or learning disability) and could they adjust them to remain within the possible range (i.e., ≥ 0).

Authors' response

We thank the reviewer for raising this important point. The 95% confidence intervals were calculated as the mean $\pm 1.96 \cdot SE$ (SE has been estimated using Jackknife approach).

In response to this feedback, we have adjusted the confidence intervals and the lower boundaries of the error bars are now ≥ 0 in Figures 3 and 5.

Also, added the following footnote text to Figures 3 and 5 legends:

"The 95% CIs were calculated as the mean $\pm 1.96 \cdot SE$ (SE has been estimated using Jackknife approach)."

5. Figure 4: all other plots show yearly estimates. Could the authors please specify on which time scale Figure 4 was measured, as it seems to be more granular than yearly measures? Is such granularity needed or would it be preferable to use the same time scale as other plots in order to aid comparability?

Authors' response

We thank the reviewer for this observation. Richness displayed in Figure 4 was measured on annual basis. The X-axis time scale has been changed to yearly from 2000 to 2013, in consistency with the other plots in the main paper and supplementary file.

6. The figures are currently very inconsistent. Could the authors please strive to make them as uniform as possible, especially were figures report the same results just for different subsets or parameters. For example:
 - a. I understand that Figures 1 (all codes) and Figure 2 (incidence codes) show the same general results but for different code sets. Could the authors harmonize/standardise these plots by using the same layout, and also report

bias-corrected results for Figure 1. This would simplify comparison of these plots.

- b. **Figure 1: Could the authors please extend the y-axis to 0 as in Figure 2?**
- c. **Figure 1: The heading “year-on-year correlations” seems misleading for 3-year correlations. Could the authors please change the heading similar to that in supplementary Figure S2?**
- d. **Figure 1: panel headers still contain underscores and all lower case. I would suggest to change headings in line with Figure 2.**
- e. **Figures 2 and S2-4: unlike Figure 1 and Figure S1, there are no red lines. Could the authors please add those lines?**

Authors' response

Thank you for these important observations on the figures. Regarding reporting the bias-corrected results for 'all codes' plots: the Jackknife bias correction was only applied for 'incident codes' due to their small count

- in comparison to 'all codes' - which may lead to biased estimates of the CCs. As suggested, we have reviewed and edited all figures in response to these comments, we extended Y-axis to 0, unified X/Y-axes scales, panel headings, added the vertical red line for year 2004 wherever is missing. We would just like to note that we have now added the correct Figure S4 as mistakenly had uploaded a duplicate of Figure 2 in the previous submission. We would hope any remaining issues to be resolved in the production stage.

- 7. **In Figure 2 there are missing values for atrial fibrillation. Could the authors please comment on why those were observed.**

Authors' response

AF was introduced in the Quality and Outcomes Framework (QOF) in the 2006-7 financial year (year 3 of the QOF) and the quality and levels of recording before that was arguably problematic enough on a few time points to avert the availability of positive-definite matrices that are essentially needed for CCA to run. CCA is based on the correlation of two data matrices (variables from group 1 and variables from group 2) and these should be numerically complete (non-missing), but as the matrix is not positive definite due the distribution of the very few used AF codes, this led to the missing values noted.

We have now acknowledged that by adding the following text to the limitations section:

"As CCA is based on the correlation of two positive-definite matrices of data that should be numerically complete, problematic quality and levels of recording as observed with AF and LD recording on some time points results in missing values as observed in e.g. Figures 1, 2 and S4."

- 8. **I would invite authors to publish their analysis code online in a repository like GitHub and link to it within their manuscript. This would greatly aid reproducibility of their results.**

Authors' response

Thank you for this useful suggestion. We have added the R code we used as an appendix (Table S2). Just to note that we have already uploaded the code lists used in this study to the online clinical codes repository to aid reproducibility of the analysis.

The following note has been added to the Methods section:

“A copy of the R code is presented in Table S2.”

Reviewer 3

Prof. Laurent Letrilliart, Département de médecine générale, E.A. 4129 « Santé, Individu, Société »

Comments to the Author:

General comments

This manuscript reports on a study exploring Read code usage for 18 mostly chronic conditions in UK general practice between 2000 and 2013. This study is original, but the methodology is not presented clearly consistently enough across the paper to allow the reader to fully understand the meaning of the objectives and of the too numerous results.

Specific comments

Introduction

- 9. The paragraph presenting the QOF should be moved just before the statement of the study objectives and linked to the issue of clinical code usage. An example of how the introduction of a new indicator in the QOF may have influenced the use of clinical codes should be provided, especially regarding type 2 diabetes.**

Authors' response

Thank you for this comment. The paragraph explaining the UK QOF scheme has been moved as suggested. The possible effect of introduced QOF indicator (mainly after the major revision in 2006) is now discussed using T2DM as an example.

Important revisions were introduced to QOF in April 2006 (covering up to March 2007) (<https://files.digital.nhs.uk/publicationimport/pub05xxx/pub05997/qof-eng-06-07-bull-rep.pdf> ; Calvert M. et al. BMJ. 2009;338:b1870) which may have potentially increased the capture of diabetes cases on that period. These revisions included: **i)** adding four new indicators and eight clinical areas (total of 16 indicators were for diabetes); **ii)** amending the existing clinical indicator sets, including diabetes; **iii)** introducing some changes to the underlying coding rules; and **iv)** redefining the diabetes register so general practitioners required to identify patients with diabetes as either having type 1 or type 2 diabetes. In a previous study (Zghebi S. et al. DOM. 2017;19(11):1537), we examined the incidence of T2DM in England, we found that incidence rates indicate that these important changes led to the capture of more new cases with T2DM in 2006 which was confirmed when we explored the calendar month of diagnosis of these cases where there was an increase in the number of new cases from April 2006 in comparison to numbers diagnosed between January-March 2006. This rise continued until March 2007 (indicating the time-window of QOF 2006-07). While we cannot observe this clearly in the code usage trends which may indicate the effect of codes used. Another example is the effect on CKD recording after the QOF revision in 2006 as described in the Discussion section (page 11).

In response to this comment, the following text has been added to the Introduction:

“Furthermore, important revisions were introduced to QOF in April 2006 (covering up to March 2007)^{26,27} including adding new indicators for diabetes, amending diabetes clinical indicator sets, and

redefining the diabetes register so general practitioners required to identify patients with diabetes as either having type 1 or type 2 diabetes, which have potentially increased the capture of diabetes cases on that period.”

Methods and results

10. Were the clinical codes under study used in the GPs' EHR for the assessment of any encounter or in the patient problem list or either?

Authors' response

Thank you for this query. There are three main clinical computer systems in use, and the structure the reviewer discusses is relevant to EMIS, as far as we are aware. We used CPRD GOLD which collects data from VISION practices where all relevant data (Read coded) are extracted into these groups of files: clinical (diagnoses and problems), referrals and immunisations. We extracted data from clinical and referral files.

11. According to the first objective, the (canonical) correlation is supposed to provide a frequency ranking, but how it does is unclear. Do the correlations measure the stability of the codes used for a particular condition? I understand that the study ranks the correlation levels of the conditions under study rather than their frequency.

Authors' response

Thank you for this comment. CCA was applied to examine the consistency of clinical code usage for a specific condition based on ranking the percentage frequency use of each clinical code by providing estimates of the correlation relationship between two groups/matrices of data (represented by Y1 and Y2). By this, CCA finds the best linear combinations maximizing the correlation between variables in group 1 and variables in group 2, where the variables are measured across a common set of units (general practices). For example, for the 2006-2007 year-on-year diabetes correlation in general practice 'P', the Y's represents the relative use of each diabetes code expressed as a percentage of the total use across all diabetes codes, where group 1 (represented by Y1) would be the percentage frequency use of each clinical code for diabetes recorded in year 2006 and group 2 (represented by Y2) would be the corresponding percentage frequency use of each clinical code for diabetes recorded in year 2007. We analysed percentage frequencies rather than frequency counts to remove any effects of variations in practice size from the estimated CCs. In R, CCs were calculated with the "Spearman" method by which the weighted linear combinations for each year are ranked across practices prior to computation of the correlation. This method produces estimates that are more robust against model misspecification.

We have modified different parts of the 'Data analysis' section and cited more references to improve the clarity of how CCA works:

"To examine the consistency of clinical code use across time, we applied canonical correlation analysis (CAA)^{31,32} to estimate 1-year (e.g. 2006 to 2007), 3-year (e.g. 2006 to 2009), and 5-yearly canonical correlations (e.g. 2006 to 2011) for code usage for each of the 18 conditions based on ranking the percentage frequency use of codes. CCA is a descriptive multivariable method that provides a measure of the canonical correlation (CC) between two groups of variables or two data matrices that should be numerically complete and non-missing."

"For example, for the 2006-2007 year-on-year diabetes correlation, the Y's represents the relative use of each diabetes code expressed as a percentage of the total use across all diabetes codes, where group 1 (represented by Y¹) would be the percentage frequency use of each clinical code for diabetes recorded in year 2006; whereas group 2 (represented by Y²) would be the corresponding percentage frequency use for each corresponding diabetes code recorded in year 2007, at the general practice level. The same applies for the 3-year and 5-year correlations."

12. Figure 2 and Figure 1 are redundant, and one of them seems enough.

Authors' response

Thank you for this comment. Figure 1 shows the correlations for all codes while Figure 2 shows the bias-corrected correlations for incident codes. We kept both figures in the main paper (representing one of the three assessed time points) to enable the comparability of the 3-year correlations of all vs. incident codes as we believe this can be of interest to the paper readers. While the 1- and 5-year

correlations are available in the supplementary data. We hope the reviewer finds this rationale satisfactory.

13. It is unclear how measures of diversity can quantify changes in clinical codes usage, as stated in the second objective.

Authors' response

Thank you for this comment. We used three measures of diversity where each show one aspect of changes in clinical code usage over time: 1) evenness measures the abundance of codes in a sample set; 2) richness measures code count i.e. the count of unique codes used, and 3) Shannon entropy index encompass both evenness and richness measures as it is based on assessing the abundance of codes and their count. We appreciate no measure is perfect, but we felt all three could shed some light into what we set out to investigate. A detailed example is outlined in our response to the comment 14 below.

14. What is the added value of using 3 indicators of diversity? The respective meaning of entropy and of evenness are difficult to understand and should be better explained.

Authors' response

Thank you for this comment. The use of these three measures provides different aspects of code usage diversity to help meet a wider range of interest in this topic as some readers may be interested in the annual count of unique codes only (richness) vs. others interested in the abundance of code usage over time (evenness). In other words, richness is known to not take abundance into account, unlike Shannon index which is a popular index of diversity and generally acknowledged it takes both abundance and count (richness) into account. The index is interchangeably referred to as 'Shannon entropy' or 'Shannon index' where the term 'entropy' just indicates the uncertainty or variability of information in a variable whose diversity is assessed by the Shannon index. The formula for calculating Shannon index is provided.

Evenness is a measure of the relative abundance of codes usage as indicated by the distribution or uniformity of codes of a particular condition used at a given year. For example, based on 100 diabetes codes records in 2006 and 100 records in 2007, if we find that the diabetes codes recorded in 2006 were based on using 4 unique codes 25 times each (similar distribution), whereas in 2007 usage was based on using a particular code 70 times (dominant code) and another code 30 times. Therefore, evenness of diabetes code usage would be higher in 2006 given the similar distribution of the codes used with no dominating code, unlike in 2007 where the codes used were less diverse and hence lower evenness.

In response to these comments, we modified the text in the Methods section and now reads as:

"First, the Shannon entropy (), an equitability and popular index of diversity. The index is interchangeably referred to as Shannon entropy or Shannon index where the term 'entropy' indicates the uncertainty or variability of information in a variable whose diversity is assessed by the Shannon index. The Shannon entropy index () was calculated as:....."

"Third, we estimated the evenness () of incident and total codes' usage, a measure of the relative usage of codes within a given year, in other words evenness will be high if all codes have a similar distribution (e.g. 100 diabetes records based on using 4 different diabetes codes, 25 times each) whereas it will be low if a few codes dominate the code usage (e.g. 100 diabetes records based on using one code 70 times and another code 30 times)."

15. It is confusing to state that the entropy index takes into account richness and evenness, since at the end this is the evenness which is calculated as a combination of the measures of entropy and richness.

Authors' response

Thank you for highlighting this point. We have dropped that sentence to improve the clarity of the text.

The new text now reads as:

“First, the Shannon entropy (), an equitability and popular index of diversity. The index is interchangeably referred to as Shannon entropy or Shannon index where the term 'entropy' indicates the uncertainty or variability of information in a variable whose diversity is assessed by the Shannon index. The Shannon entropy index () was calculated as:.....”

16. In the methods section, a possible richness measure for diabetes is 3, whereas the values presented for incident diabetes mellitus in Figure 4 are around 250. Why such an inconsistency?

Authors' response

Thank you for this comment. The hypothetical example we outlined in the Methods section (page 8) was just to simplify and help introduce the concept of the three diversity measures and how they are calculated, but it is not directly indicative of the actual results presented in the following sections. On that section, richness is equal to 3 as the illustrated example was based on three diabetes codes A, B & C.

We now clearly describe that example as hypothetical to improve clarity and the text reads as:

"To simplify what these diversity measures imply, we describe a hypothetical example:"

17. What is the need to present the incident codes and the total (prevalent) codes in some of the figures? The comparison between them is not reported in the main findings.

Authors' response

Thank you for this comment. The presentation of the results for both code types aimed to highlight the clinical code usage over time to either identify new cases of each condition (using incident codes) or new or prevalent (using total codes). We mainly comment on this comparison under the 'Clinical code usage diversity' subheading of the Results section on pages 10 & 11. The findings show a separation between the annual trends of entropy index for incident and total codes for AF, dementia, depression, CKD, and LD around 2006; also, total codes' richness indices surpass those of incident codes mainly for SMI, diabetes and cancer, whereas the evenness index of total codes exceeded that of incident codes after 2006 for most conditions unlike for depression and dementia where evenness for incident codes surpass that of total codes.

18. The red dotted line in some of the figures should be defined and probably included in all figures. I guess it represents the launching year of the QOF.

Authors' response

Thank you for this note. Correct, the red line represents the launching year of the QOF in 2004. It has now been added to all figures as suggested by the reviewer and clearly defined in the figure legends (page 22).

19. Some legends are missing on the x axes in Figure 1.

Authors' response

Thank you for this remark. We have now added missing year labels to all figures whenever needed in the main paper and supplementary file.

Discussion

20. The second paragraph of the subsection on the Comparison with previous studies (lines 9 to 33 on p12) should be related to the results of the study. In addition, the authors report the interpretation of the results of the study by Tai et al. without reporting the results of this study.

Authors' response

Thank you for this important note. We made changes to the paragraph in response to this comment and the new text reads as:

"A study by Tai et al. (2007) examined the diversity of data entry screens in four clinical computer systems available in UK general practices and assessed its impact on the variation and quality of recorded clinical data for six exemplar conditions (sore throat, tired all the time, depression, cystitis, type 2 diabetes, and myocardial infarction).²² In agreement with what we report on the large number of available codes for some conditions (high richness), Tai and colleagues found that searches for the same clinical term across the systems resulted in different results and found long code lists where the mean number of codes ranged between 12.7 and 35.2 codes per list.²² Their study concluded that the systems may contribute towards a diverse coding in primary care, suggesting the need to standardise clinical coding across systems and to adopt shorter and more restricted code lists to help improve data quality. This is an important issue for UK primary care, since the semi-structured and dynamic nature of Read codes often results in diverse and long clinical code lists. Their findings highlight the need for analyses as ours, which are lacking in current literature, investigating the real-world abundance and trends of code usage over time derived from routine clinical data."

21. It would be useful to discuss the potential of a limited reference set of clinical codes such as the Snomed CT subset (<https://dd4c.digital.nhs.uk/dd4c/publishedmetadatas/intid/971>) to improve coding consistency in general practice when Snomed CT will be implemented.

Authors' response

Thank you for this useful suggestion. One of the potentials of SNOMED CT terminology to improve coding consistency is mainly through the plan of its implementation in both primary care and secondary care systems, which will contribute to a more uniform practice unlike previously when different coding systems were used in different settings (e.g. Read codes in primary care vs. ICD codes in secondary care). This is particularly important as it has been reported that Read coding have failed in secondary care over time. ([SNOMED to replace Read Codes by 2020 https://www.digitalhealth.net/2015/10/snomed-to-replace-read-codes-by-2020/](https://www.digitalhealth.net/2015/10/snomed-to-replace-read-codes-by-2020/)) However, a possible issue with SNOMED terminology is the need for specialist browser and

reference sets, such as the general practice reference set to handle the long hierarchies of the SNOMED system. ([SNOMED to replace Read Codes by 2020 https://www.digitalhealth.net/2015/10/snomed-to-replace-read-codes-by-2020/](https://www.digitalhealth.net/2015/10/snomed-to-replace-read-codes-by-2020/)) A reference set is a mechanism that can be employed to represent value sets

of SNOMED CT components. ([Reference Set https://confluence.ihtsdotools.org/display/DOCRFSPG/2.3.+Reference+Set](https://confluence.ihtsdotools.org/display/DOCRFSPG/2.3.+Reference+Set))

To date, there is no full transition to SNOMED CT with some remaining issues in coding. More research will be needed following the complete transfer to the SNOMED CT coding to enable assessing its full capacities in improving coding consistency in primary care and potentials in overcoming the limitations observed with the Read and previous coding systems.

In response to this comment, the following text has been added to the Discussion section:

"There is potential for SNOMED CT terminology to improve coding consistency, mainly through the plan to implement it in both primary care and secondary care systems.⁴⁰ However, a possible issue with SNOMED terminology is the need for specialist browser and reference sets, such as the general practice reference set to handle the long hierarchies of SNOMED system.⁴⁰ A reference set is a mechanism that can be employed to represent value sets of SNOMED CT components.⁴¹"

22. What could be the influence of changes in the epidemiology of chronic conditions over the study period on the use of some codes? For example, the prevalence of type 2 diabetes has increased in the recent decades. Can this evolution have contributed to the changing way of coding this condition?

Authors' response

Thank you for this important comment.

Actually, we have commented on the possible relationship between clinical code selection and epidemiology in one of our past studies examining the incidence and prevalence of T2DM in England between 2004- 2014. (Zghebi et al. DOM. 2017;19(11):1537) This relationship was first highlighted by a UK study by Tate and colleagues (Tate et al. BMJ Open. 2017;7: e012905) who assessed if the quality of recording of diabetes affected diabetes incidence and prevalence rates between 1995-2014 and concluded that, in contrast to previous reports, the incidence of diabetes, based on diagnostic codes, has not increased in the UK since 2004, suggesting that the choice of codes has a significant effect on incidence estimates. Our reported T2DM incidence trends (Zghebi et al. DOM. 2017;19(11):1537) agreed with Tate's conclusion which probably reflects our approach in case definition strategy on that study by excluding type 1 diabetes cases which aimed to minimize misclassification of T2DM. These reports demonstrate a possible relationship as the reviewer rightfully stated. From GPs stance, one thing that may have changed in recent years is the coding of people being at "High risk of diabetes mellitus" and it is something that GPs are increasingly aware of (i.e. people with HbA1c 42-47 mmol/mol) that reflects changes in coding practice as a result from the evolution in diabetes prevalence in the recent years.

The following text has been added to the discussion in response to this comment:

"A possible relationship between clinical code selection and epidemiology of chronic conditions has been reported previously, for example, for diabetes.^{19,42} From general practitioners (GPs) stance, one thing that may have changed in recent years is the coding of people being at "High risk of diabetes mellitus" and it is something that GPs are increasingly aware of (i.e. people with HbA1c 42-47 mmol/mol)".

Conclusion

23. The authors conclude that "the changes were higher over longer time windows". Was it not expected or even obvious? This should probably be formulated differently.

Authors' response

Thank you for this comment. We changed the text and now reads as:

"The code usage in UK primary care was overall stable for most of the examined chronic conditions managed in general practice between 2000 and 2013, but, as would be expected, the changes were higher over longer time windows."

24. The opposition in the following sentence is not clear enough: "Diabetes, cancer, and SMI codes need to be defined carefully but SMI codes can last up to 5 years,...".

Authors' response

Thank you for this comment. We revised the text for more clarity and it now reads as:

"Diabetes, cancer, and SMI code lists have high richness and need to be defined carefully by researchers and/or clinicians which might be considerably time-consuming, but once defined SMI codes can last up to 5 years, while diabetes and cancer codes only for 2 years."